# Critical Care in SARS-CoV-2 Infected Pregnant Women: A Prospective Multicenter Study

**DOI:** 10.3390/biomedicines10020475

**Published:** 2022-02-17

**Authors:** Ana Álvarez Bartolomé, Nadia Akram Abdallah Kassab, Sara Cruz Melguizo, María Luisa de la Cruz Conty, Laura Forcen Acebal, Alejandra Abascal Saiz, Pilar Pintado Recarte, Alicia Martinez Varea, Lucas Cerrillos Gonzalez, Javier García Fernández, Oscar Martínez Pérez

**Affiliations:** 1Department of Anesthesia & Critical Care, Puerta de Hierro University Hospital of Majadahonda, 28222 Madrid, Spain; nadiaabdallah90@gmail.com; 2Department of Gynecology and Obstetrics, Puerta de Hierro University Hospital of Majadahonda, 28222 Madrid, Spain; saracruz.gine@yahoo.es; 3Fundación de Investigación Biomédica, Puerta de Hierro University Hospital of Majadahonda, 28222 Madrid, Spain; farmcruz@gmail.com; 4Department of Gynecology and Obstetrics, University Hospital 12 de Octubre, 28041 Madrid, Spain; lauratrona@gmail.com; 5Department of Gynecology and Obstetrics, La Paz University Hospital, 28046 Madrid, Spain; alejandra_as@hotmail.com; 6Department of Gynecology and Obstetrics, Gregorio Marañón University Hospital, 28007 Madrid, Spain; ppintadorec@yahoo.es; 7Department of Gynecology and Obstetrics, La Fe University and Polytechnic Hospital, 46026 Valencia, Spain; martinez.alicia.v@gmail.com; 8Department of Gynecology and Obstetrics, Virgen del Rocío University Hospital, 41013 Sevilla, Spain; lcerrillog@sego.es; 9Chairman of Anesthesia & Critical Care Department, Puerta de Hierro University Hospital of Majadahonda, 28222 Madrid, Spain; ventilacionanestesia@gmail.com; 10Maternal-Fetal Medicine Unit, Department of Gynecology and Obstetrics, Puerta de Hierro University Hospital of Majadahonda, 28222 Madrid, Spain; oscarmartinezgine@gmail.com

**Keywords:** coronavirus disease 2019 (COVID-19), severe acute respiratory syndrome coronavirus 2 (SARS-CoV-2), pneumonia, pregnancy, intensive care

## Abstract

Evidence suggests that pregnant women are at a higher risk of complications compared to the general population when infected with severe acute respiratory syndrome coronavirus 2 (SARS-CoV-2) and the reasons that lead them to need intensive care are not clear. This is a prospective multicenter study of SARS-CoV-2 positive pregnant women, registered by the Spanish Obstetric Emergency Group, with the objective to define the characteristics of the mothers who were admitted to the Intensive Care Unit (ICU) and to investigate the causes and risk factors for ICU admission. A total of 1347 infected pregnant women were registered and analyzed, of whom, 35 (2.6%) were admitted to the ICU. No differences in maternal characteristics or comorbidities were observed between ICU and non-ICU patients, except for in vitro fertilization and multiple pregnancies. The main causes of admission to the ICU were non-obstetric causes (worsening of the maternal condition and respiratory failure due to SARS-CoV-2 pneumonia, 40%) and a combination of coronavirus disease 2019 (COVID-19) symptoms and obstetrical complications (31.4%). The multivariable logistic analysis confirmed a higher risk of ICU admission when pre-eclampsia or hemorrhagic events coexist with pneumonia. The incidence of thromboembolic events and disseminated intravascular coagulation were also significantly higher among patients admitted to the ICU. Therefore, surveillance and rapid intervention should be intensified in SARS-CoV-2 infected pregnant women with the mentioned risk factors and complications. Emphasis should always be placed on anticoagulant therapy in these patients due to the increased thromboembolic risk, C-section surgery and immobilization in the ICU.

## 1. Introduction

Severe acute respiratory syndrome due to coronavirus 2 (SARS-CoV-2) was declared as a pandemic by the World Health Organization (WHO) in March 2020 and has caused more than 164 million infections worldwide and approximately 3.4 million deaths, becoming a global health emergency [1].

There has been no data to confirm that pregnant women may have a greater predisposition to infection; however, evidence suggests that SARS-CoV-2 infected pregnant women are at higher risk of hospitalization, admission to Intensive Care Unit (ICU), invasive mechanical ventilation (IMV) and extracorporeal membrane oxygenation (ECMO) therapy compared to the general population [2].

Due to the physiological changes that occur during pregnancy, women are more vulnerable to respiratory infections and associated complications, with respiratory failure being a common cause of ICU admission [3], as the increased oxygen consumption and the decreased functional residual pulmonary capacity rapidly cause hypoxemia in case of respiratory failure [3].

Although most of the coronavirus disease 2019 (COVID-19) cases in pregnant women are mild-moderate (88.5%), some women develop severe disease requiring oxygen therapy (9.8%), and up to 1.6% develop critical illness [4]. Compared to non-infected pregnant women, a higher incidence of preterm delivery, cesarean section, pre-eclampsia and other morbidities have been observed in SARS-CoV-2 infected pregnancies [4].

There also seems to be an increased risk of ICU admission compared to non-pregnant women of reproductive age [5,6], but the main causes of ICU admission are unknown, as well as the maternal and fetal complications and the evolution of these patients.

The objective of the present study was to describe the characteristics of SARS-CoV-2 infected pregnant women admitted to the ICU and to investigate the causes and risk factors for ICU admission and whether obstetric pathology and type of delivery are relevant.

## 2. Materials and Methods

This is a prospective multicenter study of consecutive cases of SARS-CoV-2 infection in a cohort of pregnant women registered by the Spanish Obstetric Emergency Group in 78 hospitals [7].

A specific database was designed to record information regarding SARS-CoV-2 infection in pregnancy; the encoded data were entered by the principal researcher of each center during the enrolment period, which occurred at the time of the SARS-CoV-2 test during pregnancy and until 6 weeks after birth. We developed an analysis plan using recommended contemporary methods and followed existing guidelines for reporting our results (Supplementary Materials Table S2) [8].

We included all SARS-CoV-2 infected obstetric patients between 26 February and 5 November 2020 by testing suspicious cases admitted to the hospital due to compatible COVID-19 symptoms and by universal screening for SARS-CoV-2 infection, which was performed on all pregnant women on admission to the delivery ward (starting on 1 April 2020). SARS-CoV-2 infection was diagnosed by double-sample positive polymerase chain reaction (PCR) from nasopharyngeal swabs. All identified cases were included in the study, regardless of clinical signs and symptoms. Then, the infected patients were divided into two groups based on whether or not they required admission to intensive care units.

### 2.1. Ethics

All procedures were approved by the Drug Research and Clinical Research Ethics Committee of Puerta de Hierro University Hospital, Madrid, Spain, (Chairperson Prof. C. Avendaño Sola) on 23 March 2020 (protocol registration number, 55/20). Each collaborating center subsequently obtained protocol approval locally (ethics committees of the participant hospitals listed in the Supplementary Materials Table S1). The registry protocol is available on ClinicalTrials.gov (accessed on 1 November 2021), identifier: NCT04558996. Upon recruitment, mothers consented to participate in the study by either signing a document when possible, or by giving permission verbally, which was recorded in the patient’s chart in the electronic clinical recording system. Ethics committees approved the possibility of verbal consent during the first three months of the pandemic, given the contagiousness of the disease and the lack of personal protection equipment. Afterward, written consent (using a patient consent form) was collected from every patient who had previously given permission verbally.

### 2.2. Study Information

Information regarding the demographic characteristics, comorbidities and current obstetric history were extracted from the clinical and verbal history of the patient. Age and ethnicity were categorized following the classification used by the CDC (Centers for Disease Control and Prevention) [6], while definitions of obstetric conditions followed international criteria [9]. For perinatal events, we recorded gestational age at delivery, the type of delivery, the type of anesthesia if required, preterm deliveries (below 37 weeks), obstetrical complications (pre-eclampsia, hemorrhagic and thromboembolic events), stillbirth, the need for invasive ventilation and/or transfusion and maternal mortality. Neonatal data, which were recorded until 14 days postpartum, included umbilical artery pH, the need for admission in the Neonatal Intensive Care Unit and neonatal mortality.

### 2.3. Data Analysis

For descriptive data, absolute and relative frequencies were used in the case of categorical variables and medians and interquartile ranges (IQR) in the case of quantitative variables. The possible association of maternal characteristics and perinatal events with ICU admission of patients were analyzed using the Pearson’s Chi-square test or Fisher’s exact test for categorical variables and the Mann–Whitney U test for quantitative variables (after checking the absence of normality of the data using the Kolmogorov–Smirnov test). Statistical tests were two-sided and were performed with SPSS V.20 (IBM Inc., Chicago, IL, USA); statistically significant associations were considered to exist when the *p*-value was less than 0.05.

To check the association of the perinatal complications that were statistically significant in the univariable analysis with ICU admission, the influence of confounding factors (and interactions) was controlled for with multivariable logistic regression modeling to derive adjusted odds ratios (aOR) with 95% confidence intervals (95% CI). Models were built for each perinatal complication separately, incorporating a range of variables and/or interactions after verifying the statistical association of potential confounding factors with ICU admission and the perinatal complication of interest (excluding intermediate variables of the causal chain) and in accordance with the ten-to-one event per variable rule to avoid model overfitting [10]. Modeling was conducted after excluding pregnancies with missing data and using a bootstrapping procedure for resampling cases (with a number of bootstrap samples equal to 999). Once the maximum multivariable logistic regression model was constructed, and to achieve the final estimated model, a confounder remained in the model if the coefficient for the perinatal complication changed by more than ten percent when the potential confounder was removed. Regression analyses were performed with the lme4 package in R, version 3.4 (RCore Team, 2017) [11]. A complete list of the final set of covariates is provided with each model in the results section.

## 3. Results

During the study period, 1347 infected pregnant women were registered and analyzed in the 78 hospitals; 35 (2.6%) were admitted to the ICU, and 1312 (97.4%) did not require intensive care (Figure 1). Of pregnant women admitted to the ICU, 91.4% were in the third trimester of gestation and only 8.6% in their second trimester; 80% of patients were admitted to the ICU after giving birth.

The answers to the following questions allow us to describe the results of our study.

### 3.1. What Are the Characteristics of SARS-CoV-2 Infected Pregnant Women Admitted to the ICU?

Among our SARS-CoV-2 infected cohort, 51.1% of the patients were asymptomatic, and 48.9% had COVID-19 symptoms (Table 1). The proportion of symptomatic patients increased up to 88.6% (31/35) in the ICU group compared to 47.9% among patients not admitted to the ICU *(p <* 0.001). Furthermore, 71% of symptomatic patients admitted in the ICU developed pneumonia (vs. 27.1% of symptomatic patients who did not need intensive care, *p* < 0.001, Table 1). A lower gestational age at the diagnosis of SARS-CoV-2 infection was observed among ICU patients (median: 34 weeks + 5 days, compared to 38 weeks + 1 day in non-ICU patients, *p* < 0.001) as well as a higher proportion of women who had required in vitro fertilization (17.1% vs. 5.2% of non-ICU patients, *p* = 0.010) and a higher rate of multiple pregnancies (11.4% vs. 1.6% of non-ICU patients, *p* = 0.003). We did not observe any other differences between groups in relation to maternal characteristics, comorbidities or current obstetric history.

### 3.2. Does the Type of Delivery Influence the Risk of Icu Admission?

We observed that the cesarean section was much more frequent in the ICU group (88.6% vs. 26.1%, *p* < 0.001), and its main reason was maternal respiratory worsening due to SARS-CoV-2 (51.4% of cases). We found that 77.4% (24/31) of C-sections performed in ICU patients were carried out before their admission to this unit and 22.6% (7/31) after they were already receiving intensive care.

Higher rates of prematurity were noticed among patients admitted to the ICU (65.7% vs. 9.6%, *p* < 0.001, Table 2), with 91.3% being preterm deliveries in ICU patients iatrogenic and only 8.7% had a spontaneous onset.

### 3.3. What Are the Main Causes of ICU Admission?

We observed that only 11.4% of the ICU admissions were for purely obstetric reasons (two cases of postpartum hemorrhages and two of severe pre-eclampsia/eclampsia in asymptomatic patients), while 40% were due to non-obstetric causes (worsening of the maternal condition and respiratory failure due to SARS-CoV-2 pneumonia) (Table 3). Additionally, 31.4% of ICU admission can be attributed to a combination of COVID-19 symptoms and obstetrical complications (pre-eclampsia and/or postpartum hemorrhages), and the remaining 8.6% due to a combination of COVID-19 symptoms and pulmonary embolism (PE) (medical complication) (Table 3).

### 3.4. Is there an Influence of Obstetrical Conditions on the Risk of ICU Admission? What Are the Most Common Maternal Complications?

The proportion of women who developed pre-eclampsia was significantly higher in the group of patients admitted in the UCI (37.1% vs. 4.3%, *p* < 0.001, Table 2); 76.9% (10/13) of pre-eclampsia cases among ICU patients were severe (including two cases of HELLP and one case of eclampsia).

On the other hand, the odds of hemorrhagic events (postpartum hemorrhages and/or abruptio placentae) among ICU patients was nearly five times higher than in non-ICU patients (20.0% vs. 4.8%, *p* = 0.002, Table 2); therefore, an increased rate of transfusions was also observed in the ICU group (26.7% vs. 0.7% in the non-ICU group, *p* < 0.001, Table 2).

However, the multivariable analysis results from the estimated models showed that pre-eclampsia, as well as hemorrhagic events by themselves, were not a risk factor for ICU admission in SARS-CoV-2 infected pregnant women, but when they coexist with pneumonia, the aOR of requiring intensive care rises to 800.0 (95% CI 125.7–1.0 × 10^10^) and to 106.2 (95% CI 17.4–7.7 × 10^8^), respectively (Table 4). In addition, as observed in both multivariable models, the presence of pneumonia by itself also represents an important risk factor for intensive care requirements (Table 4).

The incidence of thromboembolic events was also significantly higher in the group of patients admitted in the ICU (Table 2), having been affected the 5.7%, 14.3% and 8.6% of these patients by deep venous thrombosis, pulmonary embolism and disseminated intravascular coagulation, respectively, even when treated with LMWH at prophylactic doses.

### 3.5. Maternal Mortality, Stillbirth and Neonatal Outcomes

Maternal mortality in the entire series was 0.14% (*n* = 2), both cases associated with Disseminated Intravascular Coagulation (DIC) (mortality associated with DIC = 2/4 = 50%). The first patient died in the operating room because of massive postpartum hemorrhages after abruptio placentae associated with DIC. The second patient died after several days in the ICU due to severe pneumonia with acute respiratory distress syndrome associated with septic shock and the development of DIC, which led to a massive bilateral PE and cardiac arrest.

The incidence of stillbirth and neonatal outcomes are recorded in Table 2.

## 4. Discussion

To our knowledge, this is one of the first studies targeting ICU admission factors of SARS-CoV-2 infected mothers. The main strength of the study is the large cohort of SARS-CoV-2 positive deliveries (1347 from 78 centers across Spain) and the considerable quantity of ICU admissions (35) included.

The incidence of admission to the ICU in this series was 2.6%, lower than the one reported by previous studies [2,12]. This difference may be related to the PCR universal screening (regardless of mother’s symptomatology), established in the participating hospitals and also to the existence of a universal public health care system in our country, which means that every woman, regardless of their socio-economic circumstances, receive the same medical care and follow-up of pregnancy and that patients can consult and be assessed before a severe disease is developed. Finally, the long period of data collection may have resulted in better knowledge of the disease and, consequently, a decrease in the rate of ICU admissions in the second and following waves of the pandemic.

No differences in maternal characteristics or comorbidities were observed between ICU and non-ICU patients, except for IVF and multiple pregnancies (higher rates among ICU). Prior to the COVID-19 pandemic, these conditions had already been described as risk factors for obstetric morbidity and, therefore, for ICU admission, due to a higher presence of comorbidities (hypertension, thrombophilia, etc.) and complications among IVF mothers [13,14,15,16] and multiple pregnancies [17]. In a study carried out in SARS-CoV-2 infected pregnant women, a higher risk of ICU admission (due to pre-eclampsia) was reported in IVF mothers [18], as hypertension is a risk factor of a worse COVID-19 prognosis [19] and a synergistic effect of COVID-19 and pre-eclampsia cannot be ruled out nor a COVID-19 induced pre-eclampsia-like syndrome [20,21].

The high incidence of C-sections and preterm deliveries (mostly iatrogenic) that was observed in the ICU group could be explained by the urgency to terminate the pregnancy due to a worsening of the mother’s condition and respiratory failure, which was the most frequent complication described in ICU patients. We must always bear in mind that maternal oxygen consumption increases 20% during pregnancy owing to increased metabolic demands, and this, combined with a reduced functional residual pulmonary capacity, result in rapid desaturation during respiratory compromise [22]. In one of our first publications [23] with a small sample size, it was found that C-sections could be a risk factor for ICU admission; however, our new data suggest that this type of delivery might be a consequence of mother worsening, and not a risk factor.

In our SARS-CoV-2 infected cohort, the main risk factor for ICU admission is the development of pneumonia. On the other hand, when pneumonia coexists with pre-eclampsia, the risk of requiring intensive care increases exponentially. Although SARS-CoV-2 infection primarily affects the respiratory system, its systemic manifestations, such as hypertension, renal disease, thrombocytopenia and liver damage, are also found and easily confused with pre-eclampsia [24]. An increased incidence of severe pre-eclampsia has been observed in pregnant women with SARS-CoV-2 compared to non-infected [25]. However, it must be noted that the analytical signs (inflammatory, hypertensive and biochemical alterations) of COVID-19 could be interpreted as alterations due to pre-eclampsia instead and, to correctly classify these cases (and distinguish between moderate and severe pre-eclampsia), it would be necessary to measure angiogenic factors in the blood (sFlt-1/PlGF) and perform an echo-Doppler of the uterine arteries [21].

The incidence of hemorrhagic events in our entire series was 5.2% but increased up to 20% in those who required admission to the ICU. In previous studies, there was controversy over whether SARS-CoV-2 infection is a risk factor for postpartum hemorrhage [2,25,26,27]. In our study, hemorrhagic events by themselves, or associated with COVID-19 mild-moderate symptoms, did not increase the risk of ICU admission; however, when this obstetric complication coexisted with pneumonia, the risk of requiring intensive care was especially high. Therefore, we should closely monitor blood loss and analytical alterations of women with severe pneumonia to provide early management. Additionally, it would be interesting to conduct studies that analyze this association and the presence of risk factors in these women for developing hemorrhagic events.

Another consideration is the high rates of PE (0.7%, despite even prophylactic heparin in the SARS-CoV-2 infected cohort, which is higher than the ratios reported prior to the COVID-19 pandemic (1.72 cases per 1.000 deliveries for PE and 0.3 to 3.5 cases per 1.000 deliveries for DIC) [28,29]. This high incidence of thromboembolic events (and, especially high in ICU patients) could be explained by the excessive inflammation produced by cytokine release (involved in abnormal activation of coagulation pathways and inhibition of anticoagulant vias), the increase of the vasoconstrictor angiotensin II (due to angiotensin-converting enzyme two downregulation) and the immobilization of patients in the ICU [30,31]. Mortality in patients who develop DIC is 50%, like DIC’s mortality associated with sepsis of obstetric origin (50–80%) [29].

Therefore, surveillance and rapid intervention should be intensified in SARS-CoV-2 infected pregnant women with the mentioned risk factors and complications. Similarly, emphasis should be placed on anticoagulant therapy in these patients due to the increased thromboembolic risk, C-section surgery and immobilization in the ICU.

As a limitation of this study, it should be highlighted that symptomatic patients are over-represented in our study population since not all participating hospitals had a universal antenatal screening program for SARS-CoV-2 infection (so only identified symptomatic cases) or implemented the program later. Another limitation is related to the identification and classification of pre-eclampsia cases, as angiogenic markers and echo-Doppler of the uterine arteries were not performed in most of the patients since this is not a common practice in many hospitals.

Moreover, the small number of ICU admissions (*n* = 35) may have penalized the power of analyses, proven by the wide OR’s confidence intervals obtained in the multivariable analyses, which should be interpreted cautiously.

Finally, it should be mentioned that most ICU admissions registered in this cohort took place during the first wave of the pandemic when COVID-19 was unknown, and there were no homogeneous protocols/ICU admission criteria for these patients. Furthermore, the existence of Respiratory Intermediate Care Units in some participating hospitals (depending on the healthcare level of the center) could have led to an underestimation of the number of critical patients, as there were patients about to be admitted to the ICU but who were attended in these units. Additionally, the criteria for classifying the disease as mild, moderate or severe were not correctly implemented, and the specific medical treatment for COVID-19 was also very variable, depending on the protocols of each hospital and the moment of the pandemic, since they were frequently modified based on the emergent literature. Patients hospitalized for COVID-19 were treated with different drugs (corticosteroids, lopinavir/ritonavir, azithromycin, hydroxychloroquine, interferon beta1, tocilizumab and prophylactic or therapeutic heparin), and asymptomatic or symptomatic patients without hospitalization could receive or not prophylactic heparin.

## 5. Conclusions

Respiratory failure due to SARS-CoV-2 pneumonia is the main cause of ICU admission among SARS-CoV-2 infected pregnant women. Multiple pregnancy or/and in vitro fertilization are risk factors for ICU admission, as well as pre-eclampsia and hemorrhagic events when they coexist with pneumonia. The higher incidence of C-sections and preterm deliveries among ICU patients are explained by the need to terminate the pregnancy due to a worsening of the mother’s condition.

## Figures and Tables

**Figure 1 biomedicines-10-00475-f001:**
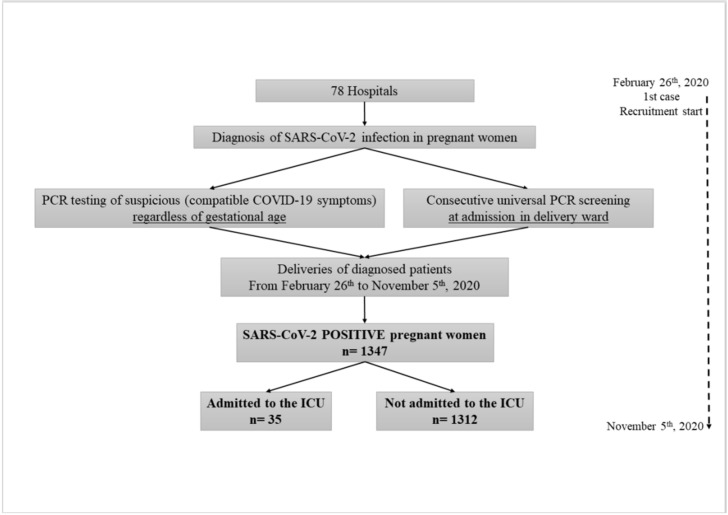
Flow chart of the study data.

**Table 1 biomedicines-10-00475-t001:** Demographic characteristics, comorbidities, current obstetric history, clinical presentation of SARS-CoV-2 infection and analytics of the study participants (n = 1347).

Number (%)	Total	Admitted ICU	No ICU	*p*-Value
1347	35 (2.6)	1312 (97.4)
Maternal characteristics	
Maternal age (years; median/IQR)	33 (28–37)	33 (26.5–38)	33 (28–37)	0.623
Age Range 18–24	183/1336 (13.7)	5 (14.3)	178/1301 (13.7)	0.858
25–34	633/1336 (47.4)	15 (42.9)	618/1301 (47.5)
35–49	520/1336 (38.9)	15 (42.9)	505/1301 (38.8)
Ethnicity White European	785/1344 (58.4)	17 (48.6)	768/1309 (58.7)	0.253
Latino Americans	374/1344 (27.8)	13 (37.1)	361/1309 (27.6)
Arab	110/1344 (8.2)	5 (14.3)	105/1309 (8.0)
Asian non-Hispanic	40/1344 (3.0)	0 (0.0)	40/1309 (3.1)
Black non-Hispanic	35/1344 (2.6)	0 (0.0)	35/1309 (2.7)
Ethnicity (2 categories) White European	785/1344 (58.4)	17 (48.6)	768/1309 (58.7)	0.232
Non-White European	559/1344 (41.6)	18 (51.4)	541/1309 (41.3)
Ethnicity (2 categories) Latino Americans	374/1344 (27.8)	13 (37.1)	361/1309 (27.6)	0.213
Non- Latino Americans	970/1344 (72.2)	22 (62.9)	948/1309 (72.4)
Blood group Type A	544/1287 (42.3)	18/34 (52.9)	526/1253 (42.0)	0.639 ^a^
Type O	535/1287 (41.6)	12/34 (35.3)	523/1253 (41.7)
Type B	154/1287 (12.0)	3/34 (8.8)	151/1253 (12.1)
Type AB	54/1287 (4.2)	1/34 (2.9)	53/1253 (4.2)
Rh Rh+	1146/1289 (88.9)	33/34 (97.1)	1113/1255 (88.7)	0.167 ^a^
Rh−	143/1289 (11.1)	1/34 (2.9)	142/1255 (11.3)
Nulliparous	516 (38.7)	15 (42.9)	501 (38.2)	0.712
Smoking ^b^	131/1290 (10.2)	1/33 (3.0)	130/1257 (10.3)	0.244
Maternal comorbidities	
Obesity (BMI > 30 kg/m^2^)	245 (18.8)	9 (26.5)	236 (18.6)	0.243
Cardiovascular comorbidities	32 (2.4)	2 (5.7)	30 (2.3)	0.204
Pulmonary comorbidities	53 (3.9)	3 (8.6)	50 (3.8)	0.169
Hematologic comorbidities	48 (3.6)	2 (5.7)	46 (3.5)	0.513
Other comorbidities	54 (4.0)	0 (0.0)	54 (4.1)	0.214
Current obstetric history	
Multiple pregnancy	25 (1.9)	4 (11.4)	21 (1.6)	0.003
In Vitro Fertilization	74 (5.5)	6 (17.1)	68 (5.2)	0.010
Hemoglobin < 10 g/dL	60/1299 (4.6)	4/34 (11.8)	56/1265 (4.4)	0.068
Platelets < 100,000/µL	12/1298 (0.9)	1 (2.9)	11/1263 (0.9)	0.281
Pregnancy-induced hypertension	36/1308 (2.8)	3 (8.6)	33/1273 (2.6)	0.069
Gestational diabetes	97/1309 (7.4)	2 (5.7)	95/1274 (7.5)	1.000
Gestational age at diagnosis (weeks + days; median/IQR)	38 + 1 (33 + 6 − 39 + 5)	34 + 5 (30 + 0 − 37 + 3)	38 + 1 (34 + 0 − 39 + 6)	<0.001
Gestational age Range at diagnosis < 13 weeks (1st trimester)	21 (1.6)	1 (2.9)	20 (1.5)	0.742
13 to < 27 weeks (2nd trimester)	149 (11.1)	3 (8.6)	
≥27 weeks (3rd trimester)		146 (11.1)
Clinical presentation of SARS-CoV-2 infection	1177 (87.4)	31 (88.6)	1146 (87.3)	
Asymptomatic	688 (51.1)	4 (11.4)	684 (52.1)	<0.001
Symptomatic	659 (48.9)	31 (88.6)	628 (47.9)
Symptomatology among symptomatic patients	
Mild-moderate symptoms	467/659 (70.9)	9/31 (29.0)	458/628 (72.9)	<0.001
Fever (with or without other symptoms)	189/467 (40.5)	4/9 (44.4)	185/458 (40.4)	
Other symptoms (different from fever)	278/467 (59.5)	5/9 (55.6)	273/458 (59.6)	
Pneumonia	192/659 (29.1)	22/31 (71.0)	170/628 (27.1)	<0.001

**Table 2 biomedicines-10-00475-t002:** Perinatal characteristics and complications of the study participants (n = 1347).

Number (%)	Total	Admitted ICU	No ICU	*p*-Value
	1347	35 (2.6)	1312 (97.4)	
Perinatal characteristics				
Gestational age at delivery (weeks + days; median/IQR)	39 + 3 (38 + 2 − 40 + 3)	35 + 2 (32 + 0 − 38 + 0)	39 + 3 (38 + 3 − 40 + 3)	0.001
Gestational age Range at delivery < 28 weeks	10 (0.7)	3 (8.6)	7 (0.5)	<0.001
28 to < 32 weeks	21 (1.6)	6 (17.1)	15 (1.1)	
32 to < 37 weeks	118 (8.8)	14 (40.0)	104 (7.9)	
≥37 weeks	1198 (88.9)	12 (34.3)	1186 (90.4)	
Type of delivery Eutocic	832 (61.8)	3 (8.6)	829 (63.2)	<0.001
Instrumental	142 (10.5)	1 (2.9)	141 (10.7)	
Cesarean	373 (27.7)	31 (88.6)	342 (26.1)	
Anestesia	1141/1334 (85.8)	33/34 (97.1)	1108/1300 (85.2)	0.049
General	30/1131 (2.7)	12/33 (36.4)	18/1098 (1.6)	
Raquídea	161/1131 (14.2)	11/33 (33.3)	150/1098 (13.7)	
Epidural	886/1131 (78.3)	8/33 (24.2)	878/1098 (80.0)	
Local	17/1131 (1.5)	0/33 (0.0)	17/1098 (1.5)	<0.001
Combinada	37/1131 (3.3)	2/33 (6.1)	35/1098 (3.2)	
Preterm deliveries (<37 weeks of gest age)	149 (11.1)	23 (65.7)	126 (9.6)	<0.001
Obstetrical complications. Hemorrhagic events:	70 (5.2)	7 (20.0)	63 (4.8)	0.002
Abruptio placentae	12 (0.9)	3 (8.6)	9 (0.7)	0.003
Postpartum hemorrhage	61 (4.5)	6 (17.1)	55 (4.2)	0.004
Hemorrhagic events at term	54/70 (77.1)	3/7 (42.9)	51/63 (81.0)	
Hemorrhagic events preterm	16/70 (22.9)	4/7 (57.1)	12/63 (19.0)	0.043
Pre-eclampsia:	69 (5.1)	13 (37.1)	56 (4.3)	<0.001
Moderate pre-eclampsia	41 (3.0)	3 (8.6)	38 (2.9)	0.087
Severe pre-eclampsia	28 (2.1)	10 (28.6)	18 (1.4)	<0.001
Medical complications Thromboembolic events:				
Deep venous thrombosis	7 (0.5)	2 (5.7)	5 (0.4)	0.013
Pulmonary embolism	10 (0.7)	5 (14.3)	5 (0.4)	<0.001
Disseminated intravascular coagulation	4 (0.3)	3 (8.6)	1 (0.1)	<0.001
Invasive ventilation Transfusion	14 (1.0) 15/985 (1.5)	14 (40.0) 8/30 (26.7)	0 (0.0) 7/955 (0.7)	<0.001 <0.001
Maternal mortality	2 (0.1%)	1 (2.9)	1 (0.1)	0.051

**Table 3 biomedicines-10-00475-t003:** Main causes of ICU admission.

	Asymptomatic	COVID-19 Mild-Moderate Symptoms	Pneumonia
No obstetrical complication			14 (40.0)
Obstetrical complications			
Pre-eclampsia/eclampsia	2 (5.7)	4 (11.4)	2 (5.7)
Postpartum hemorrhage	2 (5.7)	1 (2.9)	2 (5.7)
Pre-eclampsia + Postpartum hemorrhage			2 (5.7)
Medical complications			
Pulmonary embolism		1 (2.9)	2 (5.7)
Others	2 (5.7) ^a^	1 (2.9) ^b^	

^a^ One case of pericardial effusion associated with hemodynamic instability in the third trimester (SARS-CoV-2 infection in the first trimester) and one case of pre-gestational diabetes decompensated by SARS-CoV-2 infection, with diabetic ketoacidosis. ^b^ Medical complication unknown.

**Table 4 biomedicines-10-00475-t004:** Multivariable analysis of ICU admission risk.

Initial Maximum Model	Final Estimative Model	Variables Associated with ICU Admission	*p*-Value	OR (95% CI)
ICU admission = interaction (Pre-eclampsia and COVID-19 symptoms) + maternal age + multiple pregnancy	ICU admission = interaction (Pre-eclampsia and COVID-19 symptoms)	Mild-moderate symptoms without pre-eclampsia	*NS* ^a^	
		Pneumonia without pre-eclampsia	<0.001 ^a^	30.3 (7.6 – 2.5 × 10^8^)
		Asymptomatic with pre-eclampsia	*NS* ^a^	
		Mild-moderate symptoms with pre-eclampsia	*NS* ^a^	
		Pneumonia with pre-eclampsia	<0.001 ^a^	800.0 (125.7 – 1.0 × 10^10^)
ICU admission = interaction (Hemorrhagic events and COVID-19 symptoms) + maternal age + multiple pregnancy	ICU admission = interaction (Hemorrhagic events and COVID-19 symptoms)	Mild-moderate symptoms without hemorrhagic events	*NS* ^b^	
		Pneumonia without hemorrhagic events	<0.001 ^b^	24.4 (8.4 – 2.2 × 10^8^)
		Asymptomatic with hemorrhagic events	*NS* ^b^	
		Mild-moderate symptoms with hemorrhagic events	*NS* ^b^	
		Pneumonia with hemorrhagic events	<0.001 ^b^	106.2 (17.4 – 7.7 × 10^8^)

Pre-eclampsia: moderate + severe. COVID-19 symptoms: 3 categories = asymptomatic, mild-moderate symptoms and pneumonia. Maternal age: tested as numerical and categorical (3 categories, Table 1) variable in two alternative models. ^a^ Compared to basal category = asymptomatic without pre-eclampsia. Hemorrhagic events: Abruptio placentae + postpartum hemorrhage. ^b^ Compared to basal category = asymptomatic without hemorrhagic events.

## Data Availability

The data presented in this study are available on request from the corresponding author. The data are not publicly available due to the multicenter nature of the study.

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
