# Peer review of "Critical Care in SARS-CoV-2 Infected Pregnant Women: A Prospective Multicenter Study"

_biomedicines, 2022, doi:10.3390/biomedicines10020475_

Round 1

Reviewer 1 Report

The work is interested in: Critical care in SARS-CoV-2 infected pregnant women: a prospective multicenter study.
I have some suggestions to clarify some aspects of the work.
Check and follow the recommendations Transparent reporting of a multivariable prediction model for individual prognosis or diagnosis (TRIPOD): The TRIPOD statement https://www.equator-network.org/reporting-guidelines/tripod-statement/
Formulate the starting hypothesis and calculate the number of subjects needed to study it with sufficient statistical power. Indeed, the number of case-patients seems very low (only 35).
With the help of a statistician or methodologist, explain the tests and the validity of the statistical tests.
In addition to the AUC, present a calibration of your model and its optimism.
Have you tested the associations of variables other than linear, for example, with splines or a polynomial?
Confirm the estimates and their confidence intervals by bootstrapping. 
Present the distribution of the residuals of your models.

Author Response

REVIEWER #1

AUTHOR´S RESPONSE AND CHANGES

The work is interested in: Critical care in SARS-CoV-2 infected pregnant women: a prospective multicenter study.

I have some suggestions to clarify some aspects of the work.

Check and follow the recommendations Transparent reporting of a multivariable prediction model for individual prognosis or diagnosis (TRIPOD): The TRIPOD statement https://www.equator-network.org/reporting-guidelines/tripod-statement/

Thank you very much for your review and comments, and for suggesting the TRIPOD guide.

However, we should clarify that as our study was observational (and therefore followed the STROBE statement, supplementary table 2), our intention was not to build a prediction model, but an estimation model, in order to adjust the OR for perinatal complications (preeclampsia and hemorrhagic events) that were observed to be associated with ICU admission in the univariable analysis. We have clarified this point in the data analysis section (lines 133-134, revised highlighted manuscript) as expressed in Table 4 in the results section (Initial Maximum Model and Final Estimated Model).

1

Formulate the starting hypothesis and calculate the number of subjects needed to study it with sufficient statistical power. Indeed, the number of case-patients seems very low (only 35).

 Thank you for this comment.

We are aware of the limited number of ICU patients and highlight this limitation in the discussion section (lines 359-361, revised highlighted manuscript).

There was no baseline hypothesis; the aim of the study was to investigate the causes and risk factors for ICU admission. And, being an observational study, the sample size was not calculated, but the entire cohort of SARS-CoV-2-positive mothers (registered between February and November 2020) in which these 35 ICU patients were included, was analyzed.

In addition, a post-hoc power analysis was not performed, as this procedure is not recommended (Reference: J. Neyman and E. S. Pearson (1933) On the Problem of the Most Efficient Tests of Statistical Hypotheses. Philosophical Transactions of the Royal Society of London. Series A, Containing Papers of a Mathematical or Physical Character Vol. 231: 289-337).

2

With the help of a statistician or methodologist, explain the tests and the validity of the statistical tests.

In addition to the AUC, present a calibration of your model and its optimism.

Thank you.

The statistical tests used were already explained in the materials and methods section (lines 113-121, revised highlighted manuscript).

Fisher's exact test was used for categorical variables, which is a robust test when the expected frequency in one or more cells is less than 5 (which may occur due to the limited number of ICU patients), and nonparametric tests such as the Mann-Whitney U test were used for quantitative variables.

For comparisons spanning 2xk tables (as in the case of age range or ethnicity), Pearson's Chi-square test was used, but results were checked afterwards by pairwise comparisons with Fisher's exact test.

The tests described above are robust and appropriate when one the comparison groups has a limited sample size.

On the other hand, and as mentioned before, our multivariable models are not for prediction, but for adjusting the OR. We did not build a single model because we were aware of the limited number of events for the characteristic of interest (ICU admissions) and in order to follow the ten-to-one rule of events per variable to avoid overfitting the model (Reference: Peduzzi, P.; Concato, J.; Kemper, E.; Holford, T.R.; Feinstein, A.R. A simulation study of the number of events per variable in logistic regression analysis. J. Clin. Epidemiol. 1996, 49, 1373–1379).

3

Have you tested the associations of variables other than linear, for example, with splines or a polynomial?

Thank you.

4

Confirm the estimates and their confidence intervals by bootstrapping.

Present the distribution of the residuals of your models.

Thank you very much for this comment.

We checked the results of the models by running the analysis using the bootstrapping procedure (described in the data analysis section, lines 131-133, revised highlighted manuscript). The results changed slightly (lines 194-224) and Table 3, revised highlighted manuscript), apart from the fact that "multiple pregnancy" was left out of both models, but the aOR and confidence intervals for the maintained variables (in particular, for the categories statistically associated with ICU admission) remain very high and wide, respectively. For this reason, we have added a comment at the end of the discussion section (line 327, revised highlighted manuscript).

Reviewer 2 Report

The study is a multi-centric prospective one on the very present problem of COVID-19 and pregnancy.

I think the design and the statistical soundness of the article is of good quality.

Some minor observations though, that could help the authors for revision:

  • the differentiation of the patients depending on the severity of the disease could have further more emphasise the medical difficulties related to the management of the SARS-Cov 2 infected pregnancies
  • some information on the national standards of care, which could be different from country to country, for example, as to how asymptomatic patients are handled
  • some discussions regarding the incidence of the obstetrical situation in non-infected patients to assess some specific pathologies and medical treatments (for example, rate of cesarean section)

Author Response

REVIEWER #2

AUTHOR´S RESPONSE AND CHANGES

The study is a multi-centric prospective one on the very present problem of COVID-19 and pregnancy.

I think the design and the statistical soundness of the article is of good quality.

Thank you very much for your interesting review and comments.

1

The differentiation of the patients depending on the severity of the disease could have further more emphasise the medical difficulties related to the management of the SARS-Cov 2 infected pregnancies.

Thank you for your comment.

Being a multicenter study developed during the first and second waves of COVID-19 in Spain, the criteria for classifying the disease as mild, moderate or severe were not correctly implemented, making difficult the manage these patients according to the severity criteria. Added at the article 322-323.

2

Some information on the national standards of care, which could be different from country to country, for example, as to how asymptomatic patients are handled.

3

Some discussions regarding the incidence of the obstetrical situation in non-infected patients to assess some specific pathologies and medical treatments (for example, rate of cesarean section)

Round 2

Reviewer 1 Report

The authors have responded well to the various questions and suggestions and the manuscript is much improved. I have no other comments.